# Effect of ZEB1 Associated with microRNAs on Tumor Stem Cells in Head and Neck Cancer

**DOI:** 10.3390/ijms24065916

**Published:** 2023-03-21

**Authors:** Letícia Antunes Muniz Ferreira, Maria Antonia dos Santos Bezerra, Rosa Sayoko Kawasaki-Oyama, Glaucia Maria de Mendonça Fernandes, Márcia Maria Urbanin Castanhole-Nunes, Vilson Serafim Junior, Rogério Moraes Castilho, Érika Cristina Pavarino, José Victor Maniglia, Eny Maria Goloni-Bertollo

**Affiliations:** 1Genetics and Molecular Biology Research Unit (UPGEM), Medical School of São José do Rio Preto (FAMERP), São José do Rio Preto 15090-000, São Paulo, Brazil; 2Laboratory of Epithelial Biology, Department of Periodontics and Oral Medicine, University of Michigan School of Dentistry, Ann Arbor, MI 48109, USA; 3Department of Otolaryngology and Head and Neck Surgery, Medical School of São José do Rio Preto (FAMERP), São José do Rio Preto 15090-000, São Paulo, Brazil

**Keywords:** cancer stem cell, epithelial cell, mesenchymal transition, microRNA targeting, ZEB1

## Abstract

Cancer biologists have focused on studying cancer stem cells (CSCs) because of their ability to self-renew and recapitulate tumor heterogeneity, which increases their resistance to chemotherapy and is associated with cancer relapse. Here, we used two approaches to isolate CSCs: the first involved the metabolic enzyme aldehyde dehydrogenase ALDH, and the second involved the three cell surface markers CD44, CD117, and CD133. ALDH cells showed a higher zinc finger E-box binding homeobox 1 (ZEB1) microRNA (miRNA) expression than CD44/CD117/133 triple-positive cells, which overexpressed miRNA 200c-3p: a well-known microRNA ZEB1 inhibitor. We found that ZEB1 inhibition was driven by miR-101-3p, miR-139-5p, miR-144-3p, miR-199b-5p, and miR-200c-3p and that the FaDu Cell Line inhibition occurred at the mRNA level, whereas HN13 did not affect mRNA expression but decreased protein levels. Furthermore, we demonstrated the ability of the ZEB1 inhibitor miRNAs to modulate CSC-related genes, such as TrkB, ALDH, NANOG, and HIF1A, using transfection technology. We showed that ALDH was upregulated upon ZEB1-suppressed miRNA transfection (Mann–Whitney ** *p*_101_ = 0.009, *t*-test ** *p*_139_ = 0.009, *t*-test ** *p*_144_ = 0.002, and *t*-test *** *p*_199_ = 0.0006). Overall, our study enabled an improved understanding of the role of ZEB1-suppressed miRNAs in CSC biology.

## 1. Introduction

Cancer stem cells (CSCs) are well-known for their ability to self-renew and recapitulate the heterogeneity of primary tumors [1]. These cells are considered important targets for future therapies; therefore, the identification of their genetic signatures is crucial. Various genes, such as *CD44*, *CD117*, Prom-1 (also known as *CD133*) [2,3], *ALDH* [3,4,5], and *NANOG* [6,7], are associated with cancer stemness. Nonetheless, a better understanding of the heterogeneity of CSCs is required; therefore, we decided to use two approaches to sort CSC-enriched populations: first, using ALDH, and second, using the triple-staining of the cell surface markers CD44, CD117, and CD133 [8].

Another phenomenon that plays an important role in cancer biology is the epithelial-to-mesenchymal transition (EMT), which might be associated with CSCs in some circumstances [9]. EMT is the process of acquiring mesenchymal morphology and plays a pivotal role in cancer invasion and metastasis. Therefore, a good understanding of the potential of CSCs to undergo EMT is needed. From this standpoint, the zinc finger E-box binding homeobox (ZEB) family is among the most prominent regulators of EMT [10], and understanding its regulation by microRNAs (miRNAs) is critical. miRNAs are small RNA molecules that are capable of affecting processes that are fundamentally important for the proper functioning of an organism. Regulation by miRNAs has been widely observed in cancer studies with promising results for the prognosis, diagnosis, and treatment of diseases and to improve the quality of life of patients [11,12]. The objective of this study was to evaluate the expression of *ZEB1* and its regulation by miRNAs and genes involved in hypoxia and EMT, in addition to evaluating the potential of the two populations of tumor stem cells: ALDH+ and CD44/CD117/133 triple-positive cells.

## 2. Results

### 2.1. Analysis of In Silico and Predicted miRNAs

*In silico* analysis was performed using the Perl script in miRTarBase to search for miRNAs that regulate ZEB1. Three miRNAs were chosen for this analysis (miR-101-3p, miR-144-3p, and miR-200c-3p). Two other miRNAs, miR-139-5p and miR199b-5p, were selected from the literature, and in the bioinformatic softwares, mirDIP (microRNA Data Integration Portal), DIANA Tools-MicroT-CDS, DIANA-TarBase databases [13], and STRING v11.0 [14].

### 2.2. Analyzing ZEB1 Inhibition Using microRNAs Pre-Selected by Bioinformatic Tools

We first investigated the potential of miR-101-3p, miR-139-5p, miR-144-3p, miR-199b-5p, and miR-200c-3p to suppress ZEB1. FaDu cells showed downregulated ZEB1 mRNA expression upon transfection with miR-139-5p and miR-200c-3p mimics (Figure 1A; *, *p* = 0.02 and *, *p* = 0.01, respectively). In HN13 cells, miR-199b-5p upregulated ZEB1 expression (Figure 1B, *p* = 0.02).

Moreover, the Western blot analysis of ZEB1 protein expression in HN13 cells showed a striking reduction upon the overexpression of these miRNAs (Figure 1C).

Prediction analysis performed in DIANAmicroT-CDS also revealed the potential regulation of *NANOG* and *VEGF-A* by miR-101-3p and miR-144-3p. *HIF-1α*, *VEGF-A*, and *NTRK2* were predicted to be regulated by miR-56 and miR-+199b-5p, while miR-200c-3p was predicted to regulate the expression of *ALDH1*, *HIF-1α*, *VEGF-A*, and *NTRK2*. For miR-139-5p, no regulatory predictions were made for the selected genes (Figure 2).

Interaction analysis between the proteins expressed by the above-mentioned genes revealed that ZEB1 interacted directly with ALDH1, NANOG, HIF-1α, and VEGF-A and indirectly with NTRK2 via VEGF-A (Figure 2).

### 2.3. Expression of CSC-Related Genes upon Transfection with microRNA ZEB Regulators

While attempting to elucidate the association between EMT and CSCs, we investigated stemness mRNA gene expression upon transfection with ZEB microRNA regulators. We observed that miR-101-3p, 139-5p, and miR-144-3p increased TrkB mRNA expression (Figure 3A; *p* = 0.02, *p* = 0.01, and *p* = 0.004, respectively). Upon treatment with the miR-200c-3p mimic, the cells did not show a statistically significant difference in ALDH mRNA expression, whereas the other miRNAs upregulated its expression (Figure 3B **, *p*_101_ = 0.009, **, *p*_139_ = 0.009, **, *p*_144_ = 0.002, and ***, *p*_199_ = 0.0006). Interestingly, miR-144-3p and miR-199b-5p significantly upregulated NANOG mRNA (**, *p* = 0.002 and *, *p* = 0.02, respectively), whereas transfection with other miRNAs did not result in a statistically significant upregulation (Figure 3C). Hypoxia and its biomarker HIF-1α are associated with ALDH expression in CSCs [15,16]. Transfection with miR-144-3p upregulated the expression of HIF-1α, but no statistically significant differences were observed for the other miRNAs (Figure 3D *, *p* = 0.03). The present study sheds light on the complexity of the EMT process in CSCs.

We further assessed the corresponding protein expression by Western blot analysis, which confirmed the mRNA expression results (Figure 3E). Altogether, our findings showed that ALDH-positive cells overexpressed ZEB1 and VEGF mRNAs, while CD44/CD117/CD133 triple-positive cells upregulated miR-200c-3p, which is capable of inhibiting ZEB1 mRNA and proteins. Although these findings suggest interesting mechanisms associated with CSCs and miRNAs targeting ZEB1 expression, further studies are required to validate the ability of these cell populations to undergo EMT.

### 2.4. Isolation and Characterization of Different Head and Neck CSC Subpopulations

The presence of CSCs could be associated with poor prognosis [17]. To further investigate the relationship between CSCs and EMT, we isolated two groups of well-known CSC types based on the signatures: (1) CD44/CD117/CD133 triple-marker positive cells and (2) ALDH-positive cell populations. Cells that did not present these markers were also isolated and, hereafter, are named the control groups (Figure 4A). This was conducted using a BD FACSAria II Cell Sorter and BD FACSMelody, which can sort populations of CD44/CD117/CD133 triple-marker positive and ALDH-positive populations from head and neck cancer cells from four cultured primary cells and four immortalized cell lines (SSC28, FaDu, HN13, and HEP-2). Following the sorting, we decided to explore the properties of these populations endowed with stem-like potential. Thus, we cultivated each of these populations on an ultra-low attachment plate for 5 d, as described by Almeida et al. [18], and carried out migration and invasion assays. Overall, CD triple-marker positive and ALDH+ cells showed an enhanced potential to form spheroids, migrate, and invade, although the ALDH+ cells formed larger spheroids than CD triple-marker positive cells (Figure 4B,C), suggesting a greater stemness potential, as reported previously [18], and greater capacity for invasion and migration (Figure 5). These results highlight the central but not unique role of ALDH in CSC biology.

We found that the CD44/CD117/CD133 triple-positive population upregulated miR-200c-3p and downregulated miR-199b-5p expression (Figure 6A; *p* = 0.004 and *, *p* = 0.03, respectively). In the ALDH+ population, no significant differences were observed (Figure 6B, ns, *p* > 0.05).

However, when comparing the properties of CSCs, we observed the downregulation of ALDH mRNA in CD44/CD117/CD133 triple-marker positive cells compared to the ALDH-positive population (Figure 6C *, *p* = 0.02). We observed no difference in NANOG mRNA levels between ALDH+ and CD44/CD117/CD133 triple-positive cells (Figure 6C ns, *p* > 0.05). In CD44, no statistically significant difference was observed, whereas PROM1 (also known as CD133) was upregulated in CD44/CD117/CD133 triple-positive cells (Figure 6C, ns, *p* > 0.05; *, *p* = 0.02, respectively). Taken together, these findings indicate that these subpopulations have different transcriptional profiles.

Notably, we observed an upregulation in ZEB1 and VEGF-A mRNA expression in the ALDH+ population compared with that in CD44/CD117/CD133 triple-positive cells (Figure 6D,E ****, *p* < 0.0001 and *, *p* = 0.02). The ZEB gene family is linked to the EMT phenotype, which plays an important role in tumor progression and metastasis [19,20]. Therefore, our findings suggest that these two populations exhibit different behaviors, although further experimental validation is required to confirm this hypothesis.

## 3. Discussion

Several miRNAs have already been described as possible regulators during the development of head and neck squamous cell carcinomas, participating in cell regulation processes such as differentiation, proliferation, apoptosis, and metastasis [21]. Metastasis is directly linked to EMT, a dynamic and necessary process during embryonic development that plays an important role in cancer progression by changing the adhesion capacity and polarity of cells, which makes them invasive and migratory [22]. One important regulator of EMT is ZEB1, which plays an essential role in intracellular regulation, differentiation, proliferation, senescence, survival, and apoptosis. Its high expression is correlated with tumor development, and it has been strongly linked with intratumoral plasticity, heterogeneity, and resistance to treatment [23]; therefore, its depletion using miRNAs is a potent tool for future therapies. In several cancers, only a subpopulation of CSCs exhibit characteristics of EMT activation, which suggests that EMT in cancer cells has a strong connection with tumor stem cells [9].

Thus, our study identified new interactions involving EMT, ZEB1, and miRNAs wherein miR-139-5p and miR-200c-3p negatively regulated ZEB1 mRNA levels in the pharyngeal cancer cell line (FaDu) while miR-199b-5p upregulated them in the oral cancer cell line (HN13). This regulation illustrates two possible mechanisms of action of miRNAs: mRNA cleavage (e.g., FaDu) and translation repression would down-regulate protein levels of Zeb1 in HN13 (e.g., HN13, Figure 1C) [11], which displays individual genetic signatures for these cells. In contrast, the protein expression of ZEB1 in oral cancer cells showed a marked reduction upon transfection with these miRNAs, which confirmed their inhibitory capacity. As miRNAs may regulate more than one gene, we also observed the upregulation of TrkB and ALDH mRNAs for miR-101-3p and miRNA-139-5p, while miR-144-3p led to the upregulation of TrkB, ALDH, NANOG, and HIF-α mRNAs. miRNA-199b-5p upregulated ALDH and NANOG mRNA. Interestingly, most of the mRNAs suppressed by ZEB1 upregulated stem cell-related genes, such as TrKb, which, when activated, morphologically modifies the cells [24] and is related to metastases in colorectal and liver cancer [25], and NANOG, which upon activation expresses a phenotype similar to that of CSC and is related to poor prognosis [26]. This further indicates that EMT status and cancerous trunk association are more complex than expected. Furthermore, the miRNAs did not affect HIF-1α expression, which is consistent with the well-known role of hypoxia in stem cell maintenance [27] and the association between tumor hypoxia and CSCs, as observed in breast and head and neck cancers [15,16]. These findings indicate the possible mechanisms by which miRNAs interact with ZEB1 in the regulation of tumor stem cells.

In search of a better metric of the CSC potential, two CSC subpopulations were isolated from cell lines and primary tumors, and through qualitative assays, the triple-positive CD44/CD117/CD133 cells and ALDH+ cells presented larger spheroids in comparison with their control cells, although the ALDH + cells showed the largest spheroids and also displayed a greater capacity for migration and invasion when compared with their control cells and with triple-positive cells, which suggests less differentiated levels of stem cells, as reported in another study [18]. These results highlight the central, but not the exclusive, role of ALDH in CSC biology.

Although the lack of experimental validation for EMT status and invasiveness is a limitation of the present study, future studies addressing these points are warranted.

CD44+/CD117+/CD133+ cells deactivate EMT by downregulating ZEB1 while maintaining their potential as CSCs by upregulating NANOG expression. These results are in line with those of other studies showing that increased NANOG expression is characteristic of stem cells, including embryonic stem cells. However, its role in the induction of pluripotent stem cells is not necessary [28,29]. Triple-positive CD44/CD117/CD133 cells showed downregulated ALDH expression, which sheds light on the hypothesis that CSCs are more heterogeneous than expected and may show less differentiated levels of tumor in the trunk region [17].

Most interestingly, our findings revealed that miRNAs have a distinct expression profile in CSCs. In recent years, two subpopulations of CSCs: mesenchymal and epithelial [30,31], have emerged. Here, a higher expression of miR-144-3p and miR-200c-3p in CD44/CD117/CD133 triple-positive cells compared to that in the ALDH + population, leading to a downregulated ZEB1 expression. In summary, it was also possible to identify a downregulation in the expression of ALDH, CD44, and VEGF-A in these triple-positive cells. This relationship could be explained by the regulatory mechanisms of miRNAs, which confirm their role in tumor stem cell potential by migration, invasion, sphere formation, and ZEB1 expression assays.

## 4. Materials and Methods

This study was approved by the Research Ethics Committee of the Medical School of São José do Rio Preto (CAAE 60735316.9.0000.5415, document no. 1.814.438). Written consent was obtained from all the patients enrolled in the study. The samples were collected only after obtaining free and informed consent from patients or legal guardians. Patients who had undergone radiation and/or chemotherapy were excluded from the study.

### 4.1. Head and Neck Cancer Samples from Patients

Four fresh head and neck cancer samples were used in the primary cell culture study. To establish primary cultures, histological head and neck squamous cell carcinoma (HNSCC) was obtained from surgical resection or biopsy in the Otorhinolaryngology Service and Head and Neck Surgery, Base Hospital, São José do Rio Preto. Fresh tissue was stored in a sterile container in a transport medium composed of Dulbecco’s modified Eagle’s medium (DMEM) (Sigma, San Louis, MO, USA) and 2% antibiotic/antimycotic (AB/AM) (Invitrogen, Waltham, MA, USA) in a thermal box was transferred to the laboratory.

Upon arrival at the laboratory, HNSCC samples were immediately washed thrice with phosphate-buffered saline supplemented with 2% AB/AM. For tissue digestion, the samples were fragmented with scissors and forceps and subjected to enzymatic disintegration using 2 mL of collagenase type I (100 U) (Gibco, Waltham, MA, USA), which were then diluted in DMEM overnight at 37 °C. The cell suspension was transferred to a 15 mL centrifuge tube containing 4 mL of DMEM culture medium supplemented with 10% fetal bovine serum (FBS), 1% AB/AM, and 1% glutamine and was centrifuged at 1000 rpm for 5 min. The supernatant was discarded, and the cell pellet was resuspended in the culture medium. Table 1 shows the clinicopathological characteristics and tumor staging of the four primary cultures from the patients included in this study. Data were obtained retrospectively from medical records.

### 4.2. Cell Sorting of Head and Neck CSCs

Head and neck stem-like cells were identified by cell sorting for the surface markers CD44-phycoerythrin (PE) (BD Biosciences, San Jose, CA, USA) [33,34], CD117-fluorescein isothiocyanate (FITC) (BD Biosciences, San Jose, CA, USA), CD133-allophycocyanin (APC) (Miltenyi Biotec, Bergisch Gladbach, Germany) [2,35,36], and the intracellular marker ALDH (aldehyde dehydrogenase) [37,38,39]. ALDH activity was detected using an Aldefluor Kit (StemCell Technologies, Durham, NC, USA) according to the manufacturer’s instructions. Flow cytometry analysis was performed on FACSAria II (BD Biosciences, Mountain View, CA, USA) and FACSMelody (BD Biosciences, Mountain View, CA, USA) equipment. Table 2 summarizes all the markers used for immunophenotyping and cell sorting from primary cultures and cell lines.

### 4.3. Sphere Assay

CSCs can form spheres when cultured in an ultralow attachment plate [8]. To evaluate the ability of the sorted cells to grow in suspension as spheres, subpopulations were cultured in ultra-low attachment plates (Corning, New York, NY, USA) for 5 d [18].

### 4.4. Migration Test

Approximately 2.5 × 10^3^ cells (positive and negative) suspended in 200 µL of DMEM (Sigma, San Louis, MO, USA) were seeded in the insert on a polycarbonate filter membrane with a pore size of 8 µm in a 24-well plate (BD BioCoat migration chamber, San Josè, CA, USA). The bottom layer of the plate was supplemented with 10% FBS (Gibco, USA). The cells were incubated for 12 h at 37 °C in a humidified incubator with 5% CO_2_.

Subsequently, the bottom of the filter membrane was fixed with 4% formaldehyde for 2 min and methanol for 20 min, stained with 5% Giemsa stain and photographed under an inverted microscope at 200× magnification.

### 4.5. Invasion Test

Cells suspended in DMEM (positive and negative) were seeded at a density of 2.5 × 10^3^ on inserts containing a Matrigel layer in 24-well plates (BD BioCoat Matrigel invasion chamber, USA). The lower chambers of the plate were filled with DMEM containing 10% FBS (Gibco, USA). The cells were incubated for 12 h at 37 °C in a humidified incubator with 5% CO_2_.

The invaded cells were fixed with 4% formaldehyde for 2 min and methanol for 20 min and stained with 5% Giemsa. The cells on the upper surface of the membrane were removed with a cotton swab, and subsequently, the inserts were photographed under an inverted microscope at 200× magnification.

### 4.6. Bioinformatics Analysis

Due to the constant upgrading of the literature related to post-transcriptional regulatory control in which the miRNAs were involved, a Perl script was created in miRTarBase to search for the miRNAs that could regulate ZEB1. After the selection of ZEB1 regulatory miRNAs using DIANAmicroT v5.0 Software [13], a prediction analysis was performed to determine whether these miRNAs could regulate the genes (Nanog Homeobox NANOG), Aldehyde dehydrogenase isoform 1 (ALDH1), Vascular endothelial growth factor (VEGF-A), *Hypoxia*-*inducible factor 1*-*alpha* (HIF-1α), (Neurotrophic Receptor Tyrosine Kinase 2) NTRK2, and stem cell (CTs) marker genes which had an altered expression in the tumor microenvironment. The presence or absence of interactions between the proteins encoded by these genes was investigated using the biological database STRING v11.0 [14].

### 4.7. Cell Transfection

HN13 cells were seeded (8 × 10^4^ cells per well) in 24-well plates and transfected with synthetic miRNAs at a concentration of 10 nM with mirVana™ miRNA mimics (Thermo Fisher Scientific, Waltham, MA, USA) of miR-101-3p, miR-139-5p, miR-144-3p, miR-199b-5p, and miR-200c-3p using Lipofectamine RNAiMAX Transfection Reagent (Thermo Fisher Scientific, Waltham, MA, USA). Total RNA and proteins were collected for the assay at 48 h post-transfection. To evaluate the effectiveness of mimic transfection, miR-1 (mirVana™ miRNA mimic, miR-1 Positive Control), a *PTK9/TWF1* regulator, was used as a positive control along with its respective negative control (mirVana™ miRNA Mimic, Negative Control #1).

### 4.8. RNA Extraction

The total RNA was extracted from sorted and transfected cells using the Direct-zol^TM^ RNA Miniprep Plus isolation kit (Zymo Research, Irvine, CA, USA). RNA concentration and purity were determined using a Qubit ^®^ fluorometer (version 2.0; Thermo Fisher Scientific, Waltham, MA, USA) and the Qubit^®^ RNA High Sensitivity Assay kit, according to the manufacturer’s instructions.

### 4.9. Reverse Transcription

Complementary DNA (cDNA) was synthesized using a high-capacity cDNA Reverse Transcription Kit (Applied Biosystems, Foster City, CA, USA) in the presence of random primers according to the manufacturer’s instructions. The reverse transcription of miRNAs was performed using TaqMan^TM^ MicroRNA Reverse Transcription (Applied Biosystems, Foster City, CA, USA) with the respective primers hsa-miR-101-3p, bta-miR-139-5p, hsa-miR-144-3p, hsa-miR-199b-5p, hsa-miR-200c-3p, hsa-miR-1, RNU6B, and RNU48.

### 4.10. Quantitative Real-Time PCR

Real-time PCR was performed to quantify gene and miRNA expression using PCR Master Mix (Life Technologies, Carlsbad, CA, USA) with specific probes and the TaqMan^TM^ miRNA Assay (Thermo Fisher Scientific, Waltham, MA, USA), according to the manufacturer’s instructions. The reactions were performed in 96-well plates using StepOne Plus (Applied Biosystems) and a CFX96 Real-Time System (Bio-Rad, Hercules, CA, USA). The SDS version 2.0 and Bio-Rad CFX Manager were used to analyze the expression curve. GAPDH and RPLPO were used as internal gene controls, and RNU6B and RNU48 were used as internal miRNA controls. To quantify the relative expression of genes and miRNAs, the formula 2^−ΔΔCq^ [40] was used and normalized to that of the negative control (NC = 1.0).

### 4.11. Western Blotting

Proteins were extracted from sorted and transfected cells using sequential extraction with the Direct-zol^TM^ RNA Miniprep Plus Extraction Kit (Zymo Research, Irvine, CA, USA) and TRIzol Reagent (Applied Biosystems), according to the manufacturer’s instructions. Protein quantification was performed using the Pierce^TM^ BCA Protein Assay Kit (Thermo Fisher Scientific) on a spectrophotometer (Fisher Scientific Wellwash). The total protein (16 µg) was run in Bolt^TM^ from 4 to 12%, Bis-Tris, 1.0 mm, Mini Protein Gel, 10-well (Invitrogen, Waltham, MA, USA) and transferred to an iBlot^TM^ Transfer Stack polyvinylidene difluride (PVDF) membrane using an iBlot Gel Transfer Device (Invitrogen). The PVDF membrane was blocked in 5% Bovine Serum Albumin/Tri-buffered saline-Twen BSA/TBS-T 1× for 1 h at 71.6 °F, followed by incubation with the primary antibody overnight at 4 °C. Thereafter, the PVDF membrane was incubated with the appropriate secondary antibody conjugated to horseradish peroxidase for 1 h at 71.6 °F. The primary and secondary antibodies used in this study are listed in Table 3. The membranes were exposed using the enhanced chemiluminescence (ECL) method (Kit GE Healthcare) according to the manufacturer’s instructions.

### 4.12. Statistical Analyses

Statistical analyses were performed using GraphPad Prism v8.0.1 (Graphpad Software, San Diego, CA, USA). The relative quantification values of samples for each variable analyzed were submitted to D’Agostino and Pearson or Shapiro–Wilk tests for normal distribution. A one-sample *t*-test (parametric) and Wilcoxon’s test (non-parametric) were used to compare gene expression before and after the functional analysis. For the comparison analysis, the unpaired *t*-test with Welch’s correction was used for samples that showed a normal distribution, whereas the Mann–Whitney test was used for samples without a normal distribution.

(* *p* < 0.05, ** *p* < 0.01, *** *p* < 0.001, **** *p* < 0.0001).

## 5. Conclusions

These findings point to the crucial role of miRNAs in the regulation of tumor development, including the levels of CSC differentiation and the EMT phenotype. Therefore, our findings provide new epigenetic strategies to counteract tumor plasticity and behavior.

## Figures and Tables

**Figure 1 ijms-24-05916-f001:**
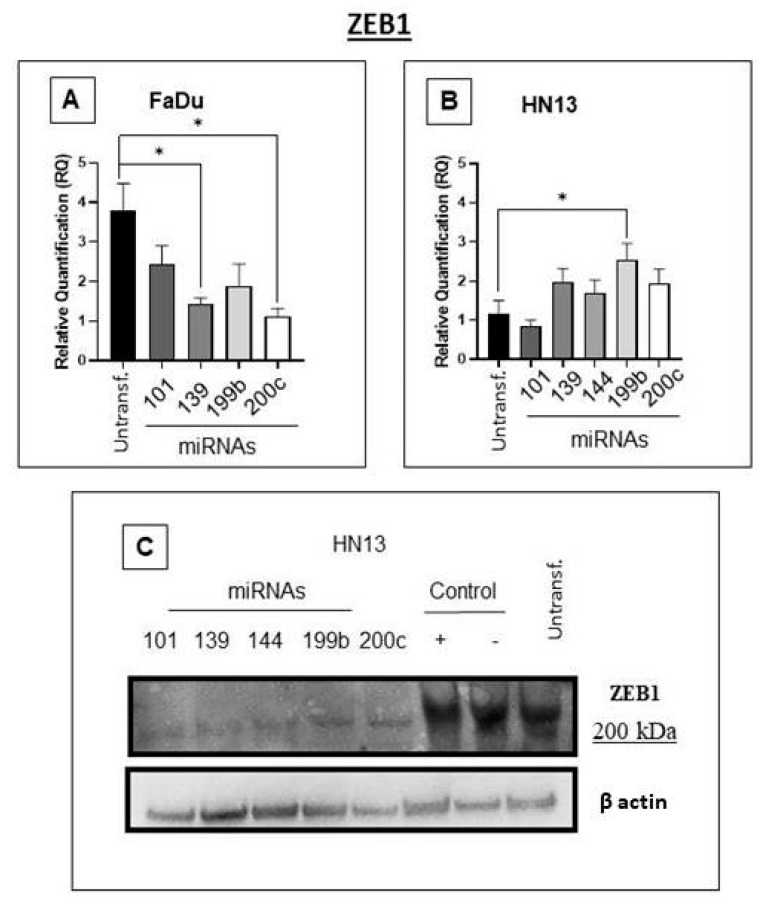
Based on bioinformatics analysis, we investigated five miRNAs as potential ZEB1 regulators (miR-101-3p, miR-139-5p, miR-144-3p, miR-199b-5p, and miR-200c-3p). (**A**) In the pharyngeal cancer cell line (FaDu), miR-139-5p and miR-200c-3p significantly downregulated ZEB1 mRNA; however, this was not observed for other miRNAs (*t*-test *, *p* = 0.02, *, *p* = 0.01, and ns, *p* > 0.05, respectively). (**B**) For the oral cancer cell line HN13, transfection with miR-199b-5p significantly upregulated ZEB1 mRNA expression, but transfection with other miRNAs showed no statistical significance (*t*-test *, *p* = 0.02, and ns, *p* > 0.05, respectively). (**C**) HN13 transfected with miRNAs (miR-101-3p, miR-139-5p, miR-144-3p, miR-199b-5p, and miR-200c-3p) and downregulated ZEB1 protein levels as observed by Western blot assay.

**Figure 2 ijms-24-05916-f002:**
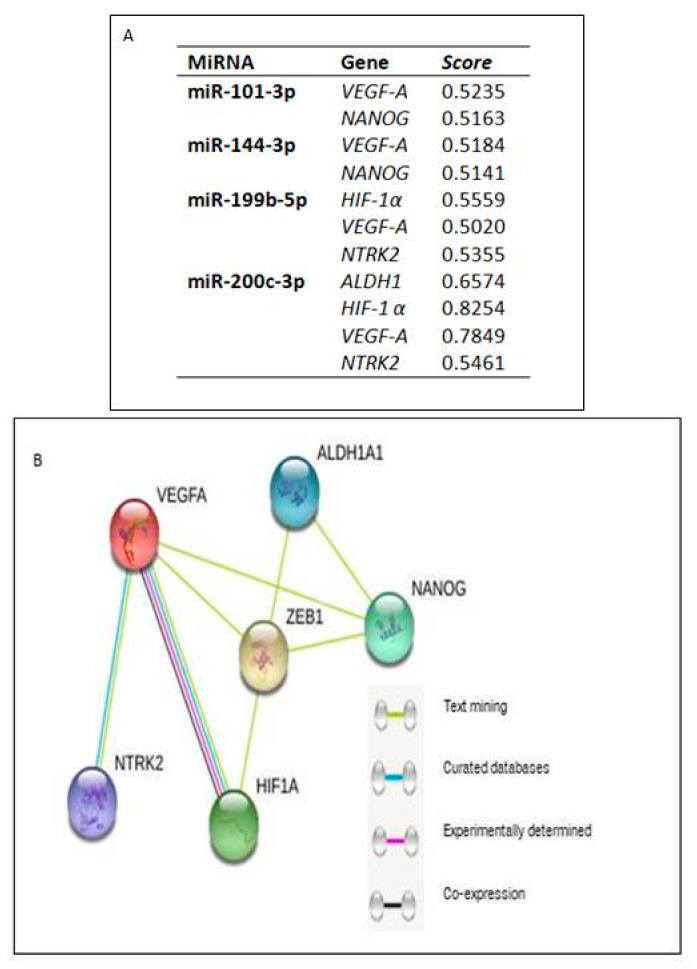
(**A**) DIANA-microT-CDS miRNA prediction analysis; considering the score greater than 0.5 (**B**) Interaction between the proteins ZEB1, NANOG, ALDH1, VEGF-A, HIF-1α, and NTRK2.

**Figure 3 ijms-24-05916-f003:**
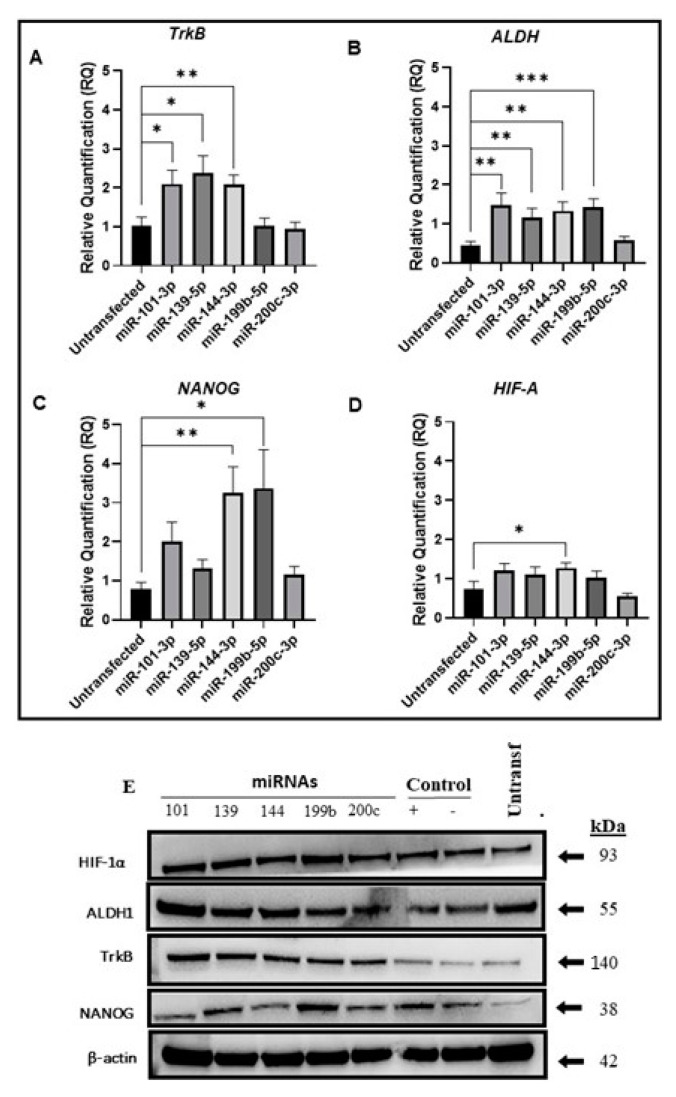
ZEB 1 suppression by miRNAs maintains cancer stem cell properties. mRNA expression levels of cancer stem cell-related genes (**A**–**D**). (**A**) Cells transfected with ZEB1 inhibitor miRNAs upregulate TrkB mRNA expression apart from miR-101-3p, miR-139-5p, and miR-144-3p (*t*-test *, *p* = 0.02, *, *p* = 0.01, and **, *p* = 0.004, respectively). (**B**) miR-200c-3p does not affect ALDH mRNA expression compared to the other miRNAs, which increase its expression (Mann–Whitney **, *p*_101_ = 0.009, *t*-test **, *p*_139_ = 0.009, *t*-test ** *p*_144_ = 0.002, and *t*-test ***, *p*_199_ = 0.0006). (**C**) NANOG mRNA is significantly upregulated after transfection with miR-144-3p and miR-199b-5p and shows no significant upregulation for the other miRNAs (*t*-test **, *p* = 0.002, and *, *p* = 0.02, respectively). (**D**) HIF-1α mRNA levels are upregulated after transfection with miR-144-3p but shows no statistical significance for the other microRNAs (*t*-test *, *p* = 0.03). (**E**) Western blot results of mRNA expression findings.

**Figure 4 ijms-24-05916-f004:**
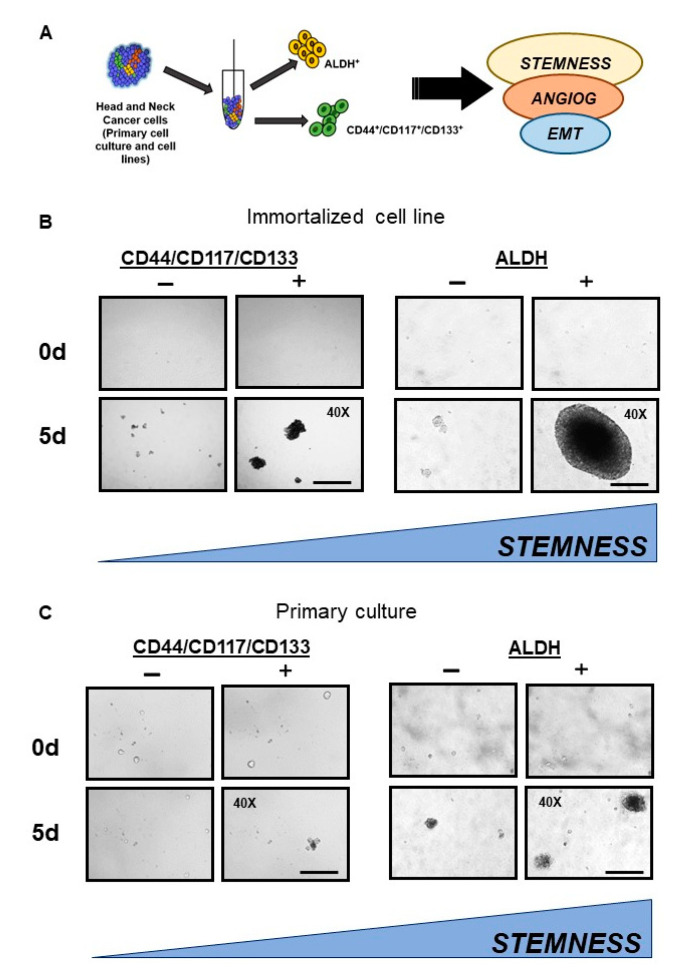
Isolation of distinctive cancer stem cell behavior. (**A**) Schematic illustration of FACS isolation of the two populations of stem cells from head and neck cancer cell lines and primary culture. DEAB negative was the basis for gating ALDH^+^ cells. Meanwhile, the ALDH^−^ and CD44^−^CD117^−^CD133^−^ cells were also sorted. (**B**) Isolated CDs^+^ and ALDH^+^ cells cultivated in ultra-low attachment conditions showed an increased number of spheroids compared with the negative control groups. Moreover, ALDH+ cells enhanced stemness potential by a different diameter in the head and neck cancer cell line (FaDu ALDH^+^ and HEP2 CDs^+^ cells). (**C**) ALDH^+^ cells show an increased number of spheroids compared to those arising from the primary cultures of head and neck squamous cell carcinoma.

**Figure 5 ijms-24-05916-f005:**
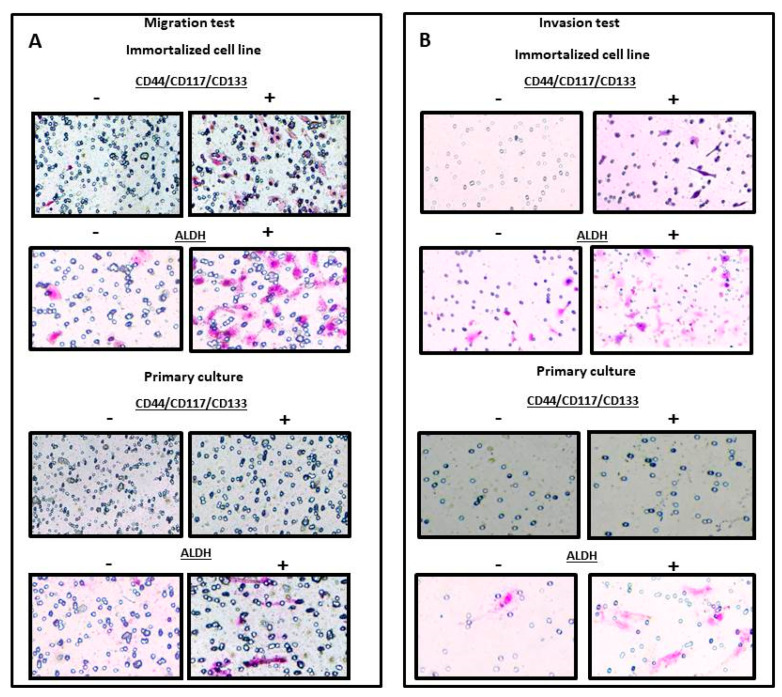
(**A**) Microphotography of the cells submitted to the migration assay (200× magnification). The isolated CD triple-marker positive and ALDH + cells showed a greater ability to migrate than the negative control group. In addition, the ALDH+ group showed a greater ability to migrate than the CDs+ group. (**A**,**B**) Microphotography of the cells subjected to the invasion assay (200× magnification). Cells labeled positively in general showed greater invasiveness than the negative control group, and a greater capacity for invasion was also observed in the ALDH+ group for the CD triple-marker positive cells.

**Figure 6 ijms-24-05916-f006:**
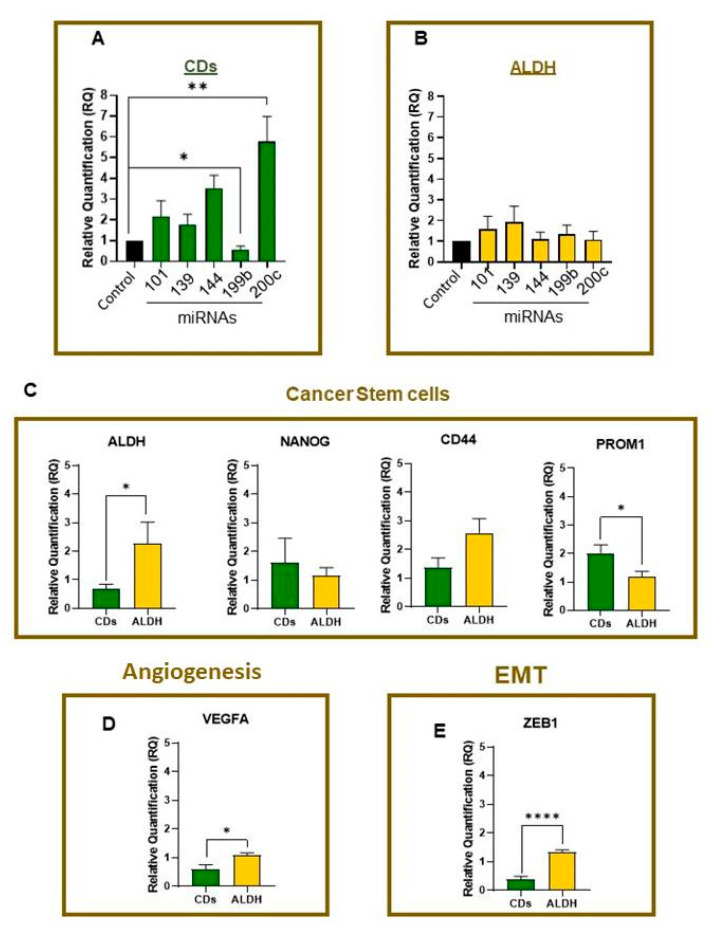
(**A**) CD44/CD117/CD133 triple-positive cell overexpresses miR-200c-3p compared with triple-negative cells (*t*-test **, *p* = 0.004) and downregulates miR-199b-5p (*t*-test *, *p* = 0.03). (**B**) We observed no difference between ALDH+ and ALDH- populations for all miRNAs (*p* < 0.05). (**C**) Cancer stem cell-related gene expression in the ALDH+ cells (Mann–Whitney test *, *p* = 0.02) was not statistically significant for NANOG (Mann–Whitney test ns, *p* > 0.05) and CD44 (*t*-test ns, *p* > 0.05), and CD44/CD117/CD133 triple-positive cells showed an augmented expression of PROM1 (*t*-test *, *p* = 0.03). The mRNA expression of the angiogenesis and EMT genes: (**D**,**E**) VEGF-A and ZEB1 increased in ALDH+ cells (*t*-test *, *p* = 0.02 and ****, *p* < 0.0001, respectively).

**Table 1 ijms-24-05916-t001:** Clinicopathological characteristics and tumor staging of samples from patients with HNSCC.

Samples	Age	Gender	Smoking	Alcohol Use	Site	TNM *
pHNC1	76	Male	Yes	No	Oral cavity	T2N0M0
pHNC2	61	Male	Yes	Yes	Larynx	T2N0M0
pHNC3	71	Male	Yes	No	Larynx	X
pHNC4	64	Male	No	No	Larynx	T1N1M0

* TNM = Tumor nodal metastasis [32].

**Table 2 ijms-24-05916-t002:** Markers used for immunophenotyping and cell sorting.

Samples and Cell Lines	Markers
pHNC1	CD44/CD117/CD133
pHNC2	CD44/CD117/CD133
HEp-2	CD44/CD117/CD133
HN13	CD44/CD117/CD133
pHNC3	ALDH
pHNC4	ALDH
FADu	ALDH
SCC-28	ALDH

**Table 3 ijms-24-05916-t003:** Antibodies used for Western blotting reactions.

Antibodies	Clone	Origin	Dilution	Manufacturer
Anti-ZEB1	ab203829 *	Rabbit	1:500	ABCAM
Anti-NANOG	# MA1-017 *	Mouse	1:1000	Invitrogen
Anti-ALDH	ab52492 *	Rabbit	1:500	ABCAM
Anti-VEGF-A	ab1316 *	Mouse	1:100	ABCAM
Anti-HIF-1α	700505 *	Rabbit	1:1000	Invitrogen
Anti-TrkB	NBP2-5 2523 *	Mouse	1:1000	Novus Biologicals
β-actina	A1978 *	Mouse	1:500	Sigma–Aldrich
Anti-mouse IgG HRP	A9044 *	Rabbit	1:10,000	Sigma–Aldrich
Anti-Rabbit IgG HRP	ab97051 *	Goat	1:10,000	ABCAM

* Catalog number.

## Data Availability

The data are maintained in the Genetics and Molecular Biology Research Unit-UPGEM, Medical School of São José do Rio Preto, FAMERP.

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
