# Peer review of "Effect of ZEB1 Associated with microRNAs on Tumor Stem Cells in Head and Neck Cancer"

_ijms, 2023, doi:10.3390/ijms24065916_

Round 1

Reviewer 1 Report

In this manuscript, authors reported that role of ZEB1 association with microRNAs on cancer stem cells of head and neck cancer. They showed that ZEB1 expression by transfections of various miRNAs in pharyngeal cancer cell line, FaDu and oral cancer cell line, HN13. Next, authors performed isolation of cancer stem cells, ALDH+ and CD44+/CD117+/CD133+ from primary cells and head and neck cancer cell lines to study the expression of stem cell related genes, EMT and angiogenesis. However, this study lacks a proper rationale, strong results, mechanistic study and characterization of sorted cancer stem cells using in vitro and in vivo approaches. The following major and minor concerns can further strengthen the quality of manuscript.

Major comments:

1. It is not clear that the rationale behind in picking up these 5 miRNAs. If it is based on bioinformatics analysis include that data.

2. Authors wants to study the association between EMT and cancer stem cells. But failed to show the EMT genes, E-cadherin, vimentin and N-cadherin expression levels and stem cell related experiments in response to ZEB1 miRNA regulators.   

3. Isolation of tissue specific cancer stem cells using ALDH, CD44, CD133 and CD117 widely studied. Though, authors needs to be characterized these sorted cancer stem cells (spheres assay only) using in vitro (migration, invasion, EMT markers expression) and in vivo mouse models (tumor progression/metastasis).

4. This study is not addressed well the detailed mechanism of regulation of ZEB1 by miRNAs and its association with cancer stem cells.

5. Rescue experiments with silencing of ZEB1 or miRNAs would be important to show its association with cancer stem cell related phenotypes.

6. Cancer stem cells plays a major role in metastasis. Therefore, authors should show the effect of modulation of ZEB1 regulated miRNAs in cancer stem cells on metastasis of head and neck cancer using in vivo mouse model.

Minor comments:

1. Authors needs to be perform extensive proofreading to correct grammar mistakes and some typo errors.

Author Response

Dear Reviewer,

The point-by-point is in the word archive, and highlighted in the manuscript text.

Sincerely,

Reviewer 2 Report

This study is interesting with clinical significance. The authors put forward a new point of view on tumor stem cells. The followings are comments to the authors.

1.Please provide the materials and methods involved in this study.

2. Please demonstrate the meanings of different color lines in Figure 5. I suggest the authors use arrows instead of lines for the interaction between the proteins ZEB1, NANOG, ALDH1, VEGF-A, HIF-1α and NTRK2.

4. How did the authors choose miR-101, miR-139, miR-144, miR-199b, and miR-200c to study?

Author Response

(The authors gave the same response as above.)

Round 2

Reviewer 1 Report

I satisfied with the revised version and recommend the present form of manuscript for publication.